# Self-Supervised Voice Denoising Network for Multi-Scenario Human–Robot Interaction

**DOI:** 10.3390/biomimetics10090603

**Published:** 2025-09-09

**Authors:** Mu Li, Wenjin Xu, Chao Zeng, Ning Wang

**Affiliations:** 1Key Laboratory of Autonomous Systems and Networked Control, Ministry of Education, College of Automation Science and Engineering, South China University of Technology, Guangzhou 510640, China; 202320116663@mail.scut.edu.cn (M.L.); wenjinxu@ieee.org (W.X.); 2Department of Computer Science, University of Liverpool, Liverpool L69 3BX, UK; chaozeng@ieee.org; 3School of Computing and Digital Technologies, Sheffield Hallam University, Sheffield S1 2NU, UK

**Keywords:** human–robot interaction, voice denoising, self-supervised learning, data synthesis

## Abstract

Human–robot interaction (HRI) via voice command has significantly advanced in recent years, with large Vision–Language–Action (VLA) models demonstrating particular promise in human–robot voice interaction. However, these systems still struggle with environmental noise contamination during voice interaction and lack a specialized denoising network for multi-speaker command isolation in an overlapping speech scenario. To overcome these challenges, we introduce a method to enhance voice command-based HRI in noisy environments, leveraging synthetic data and a self-supervised denoising network to enhance its real-world applicability. Our approach focuses on improving self-supervised network performance in denoising mixed-noise audio through training data scaling. Extensive experiments show our method outperforms existing approaches in simulation and achieves 7.5% higher accuracy than the state-of-the-art method in noisy real-world environments, enhancing voice-guided robot control.

## 1. Introduction

In recent years, with the development of Vision–Language–Action (VLA) models [1,2,3,4] facilitating significant advancements in robotic manipulation, there has been growing interest in integrating action prompts and voice interaction. However, one of the most critical and unresolved issues is ensuring effective command recognition in human–robot interaction (HRI). Current VLA models often rely on voice encoders (e.g., LLama [5], T5 [6]) to understand task prompts during HRI in real-world applications, but may miss critical instructions due to environmental noise [7]. This leads to the first research question: **(1) How can audio commands be stably transmitted in real-world interactive environments?**

Regarding the first question, supervised training strategies [8,9] have emerged as reliable tools for voice denoising during HRI. Nevertheless, this method requires collecting multiple high-quality audio samples for training, a process that demands a carefully controlled recording environment, resulting in both increased complexity and higher cost. Thanks to previous works [10,11,12], researchers have employed self-supervised learning to train models for generating voice sequences. Noise variation can also be estimated from noisy audio datasets [13] through latent audio features [14], while Large-Language Model (LLM) [15] enables audio-to-text conversion followed by semantic reasoning to interpret distorted speech. However, many of these systems still struggle in real-world robotic applications, particularly in multi-scenario or multi-speaker environments. First, existing voice datasets fail to adequately simulate ambient sound, particularly in cases of overlapping human voice, limiting practical performance. Second, interactive audio exhibits discrete properties, limiting the effectiveness of time-domain denoising for frequency feature reconstruction. Finally, LLMs are prone to hallucinations, generating inaccurate or unsafe responses that may produce erroneous robotic commands with potentially harmful consequences. This leads to two further research questions: **(2) How to scale training data for enhanced real-world denoising capability? (3) How to effectively denoise audio for real-world HRI applications?**

Through answering questions (1) to (3), a self-supervised voice denoising approach is proposed to enhance robotic audio command transmission quality in real-world applications, as illustrated in Figure 1. In contrast to previous work, our model employs realistic synthetic data (i.e., *multi-environment noise* or *overlapping voices*) in self-supervised training and integrates the convolution and attention network to extract key features from the audio magnitude and phase information. The intuition behind our design is the integration of *scaling data* and *self-supervised denoising* capabilities, where both training inputs and targets are directly generated from noisy signals. The specialized denoising architecture with regularized loss functions delivers results matching supervised training, providing reliable audio for VLAs. A comprehensive comparison of the evaluation results shows that our model outperforms state-of-the-art methods in *simulation benchmarks* and achieves a 7.5% improvement in *real-world manipulation tasks*.

The contributions of this work are summarized below:We propose training the denoising model with synthetic data, using data scaling to enhance robustness in model application.We present a self-supervised method using a convolutional network to denoise both magnitude and phase components in the frequency domain.We demonstrate proof-of-concept of the proposed approach in both simulation and hardware. For the benefit of the community, we will open-source part of the scaled dataset at https://tinyurl.com/2ybfz3jt (accessed on 25 August 2025).

## 2. Related Work

**Synthetic Data Application**. Synthetic data can rapidly generate large-scale training sets, enhancing the practicality of datasets [16,17,18]. Recent advances in voice synthesis include generating various pulses, tones, and altered timbres [19,20,21]. In contrast, HRI applications require interactive voice in diverse environments. Prior work using public audio datasets for denoising model training often shows poor generalization in complex real-world scenarios. In addition, collecting pure datasets requires significant expense. Therefore, we aim to use audio synthesis to generate diverse datasets for model training, covering voices in multiple scenarios.

**Voice Denoising Network**. Noise–Clean Training (NCT) [11,22,23,24], a common audio denoising method, uses clean audio signals as training targets. However, it struggles to handle variable environmental noise in real-world applications. Thus, noise-only training has recently gained wide attention [25,26]; the network can be trained with multiple independent noisy voice signals using self-supervised methods. Xu [14] leverages silent intervals in speech signal to isolate noise-only periods for time-varying noise estimation, and studies like [10,11] demonstrate that voice denoising models can be built from datasets containing diverse noise. Inspired by Noisy-to-Noisy, Wu [12] employs a complex-valued network architecture for multi-dimensional audio denoising and [13] constructs mixed-audio datasets combining robot ego-voice, fan noise, and concurrent human speech to improve low-frequency noise suppression. However, these methods fail to process overlapping audio in multi-speaker environments, making it difficult to extract target signals in interaction dialogues. In our work, we improve the model’s denoising capability for unseen environments by synthesizing audio from diverse scenarios and expanding the training dataset.

**Voice Control in HRI**. Voice can be used in HRI to deliver action instructions and guide robot manipulation [27,28,29,30,31]. The development of LLMs [5,6,15] has led to significant advancements in voice reasoning, particularly in environments requiring high adaptability and generation. Lai [28] proposed a natural multimodal fusion-based HRI framework (NMM-HRI) that combines voice commands and deictic postures, leveraging LLMs to generate robot action sequences. In [32], the PlanCollabNL framework combines LLMs with task planning to generate robot action subgoals from abstract natural language goals. Building on the success of VLA models [2,3,4], voice interaction is gaining significant attention for its ability to provide operational conditions and task requirements for robotic actions through voice prompts. Most VLAs [2,3,4,33,34], treat the frozen-weight LLM as a policy to generate action commands for robot operation. However, none of the works address that audio distortion in noisy environments further reduces the LLM model’s instruction recognition accuracy. In this work, we aim to use a self-supervised denoising algorithm to optimize audio quality, enabling stable transmission of audio prompts to VLAs.

## 3. Preliminaries

### 3.1. Human–Robot Voice Interaction System

In the human–robot voice interaction scenario, we consider there are two equidistant microphone arrays (i.e., simulating human binaural distribution) to capture the interactive sound in the environment, as shown in Figure 2. This ensures recorded voice signals vary randomly and nonlinearly over time. The interaction architecture uses the Z-transform to convert signals to the time–frequency domain, with the collected signal M(z) composed of the source sound T(z) and the mixing matrix A(z). The following equation holds:(1)M(z)=T(z)·A(z),(2)$$A\left(z\right)=\left[\begin{array}{cc} 1 & a_{0}z^{-d_{0}}\\ a_{1}z^{-d_{1}} & 1 \end{array}\right]\;,$$
where $d_{i}$ and $a_{i}$ denote the fractional delay in samples and the attenuation factor along the indirect signal path, respectively, and i=0,1 represents the two recording microphones. However, one challenge in voice denoising is that M(z) contains complex environmental interference, and direct application of audio enhancement can easily distort the source sound. It is valuable to separately denoise frequency-domain information (audio magnitude or phase) to enable effective extraction of key semantic information from discontinuous mixed-audio signals.

### 3.2. Self-Supervised Denoising Strategy Foundation

A clean voice dataset for training models demands a huge amount of data; Noise2Noise (N2N) [35] introduces a self-supervised optimization method to overcome the lack of clean data. It uses pairs of independent noisy data *m* and *n* (*which share the same ground-truth s*) for learning. Kashyap [11] proposed that symmetric noisy voice signals in the same environment can also serve as training pairs. Equation (Equation 3) indicates the loss function in denoising network $f_{\theta }$ with respect to parameter θ.(3)$$\underset{\theta }{\arg\;\min}\;E{∥f_{\theta }\left(m\right)-n∥}_{2}^{2}.$$

As reported in [36], denoising performance comparable to supervised training with ground truth can be achieved by minimizing Equation (Equation 4) as follows:(4)$$\underset{\theta }{\arg\;\min}\;E{∥f_{\theta }\left(m\right)-s∥}_{2}^{2}.$$

Wu [12] proposed that an audio sub-sampler *P* can be used to transform the noisy audio signal *x* into a training pair $\left(p_{1},p_{2}\right)$, i.e., $\varepsilon =E\left(p_{2}\left(n\right)\right)-E\left(p_{1}\left(n\right)\right)$, which can be directly utilized to train the denoising network:(5)$$\underset{\theta }{\arg\;\min}\;E{∥f_{\theta }\left(p_{1}\left(x\right)\right)-p_{2}\left(x\right)∥}_{2}^{2}.$$

Given an ideal denoiser $f_{\theta }^{*}$ trained on clean data and defined by Equation (Equation 5), it guarantees that the output of $f_{\theta }^{*}$ for noisy input *x* is the clean voice *y*, i.e., $y=f_{\theta }^{*}\left(x\right)$. Therefore, the optimal network is as follows:    (6)$$\begin{array}{cc} \; & E\left\{f_{0}^{*}\left(p_{1}\left(x\right)\right)-p_{2}\left(x\right)-\left(p_{1}\left(f_{0}^{*}\left(x\right)\right)-p_{2}\left(f_{0}^{*}\left(x\right)\right)\right)\right\}\\ \; & \;=p_{1}\left(y\right)-E\left\{p_{2}\left(x\right)\right\}-\left(p_{1}\left(y\right)-p_{2}\left(y\right)\right)\\ \; & \;=p_{2}\left(y\right)-E\left\{p_{2}\left(x\right)\right\}=0. \end{array}$$

Neighbor2Neighbor (Ner2Ner) [37] indicates that a regularization term μ can address the non-zero difference between $p_{1}\left(x\right)$ and $p_{2}\left(x\right)$ in θ. Inspired by [37], we add spectral constraints to minimize the gap between the output and the target audio, while [38] demonstrates that spectral structure optimization enhances the ability of the convolutional network in voice enhancement. Hence, instead of directly optimizing Equation (Equation 5), a constrained optimization problem incorporating Equation (Equation 6) and the spectral optimization loss is considered:(7)$$\begin{array}{cc} \; & E{∥f_{\theta }\left(p_{1}\left(x\right)\right)-p_{2}\left(x\right)-\left(p_{1}\left(f_{\theta }\left(x\right)\right)-p_{2}\left(f_{\theta }\left(x\right)\right)\right)∥}_{2}^{2}\\ \; & \;+\underset{\theta }{\arg\min}\;E{∥f_{\theta }\left(p_{1}\left(x\right)\right)-p_{2}\left(x\right)∥}_{2}^{2}\\ \; & \;+\delta E{∥\log\left(S\left(f_{\theta }\left(p_{1}\left(x\right)\right)\right)\right)-\log\left(S\left(s_{2}\left(x\right)\right)\right)∥}_{2}^{2}, \end{array}$$
where $S(f_{\theta }\left(p_{1}\left(x\right)\right))$ represents the optimized audio spectrum, and $S(s_{2}\left(x\right))$ represents the target audio spectrum.

## 4. Methodology

**Project Overview**. The proposed framework comprises three main components: **(1)** synthetic voice generation for training data expansion (Section 4.1); **(2)** a self-supervised denoising framework for environmental noise suppression (Section 4.2, Section 4.3, Section 4.4 and Section 4.5); **(3)** real-world deployment of denoising model in human–robot interaction scenarios (Section 4.6). The overall workflow of the approach is depicted in Figure 3.

**Problem Formulation**. The voice denoising model can endow a robot with the ability to recognize commands in *noisy and audio-overlapping* environments. This enables the robot to perform various tasks by leveraging a specific level of environmental perception. Formally, our goal is to enhance the ability of the denoising model to extract and understand conversational information in complex environments.

**Model Architecture**. As illustrated in Figure 4, a noisy voice signal *x* is passed through a sub-sampler to generate a noisy training data pair $\left(p_{1}\left(x\right),p_{2}\left(x\right)\right)$, sharing the same ground truth. We transform $p_{1}\left(x\right)$ into STFT representations as input to the training network, and design the encoding and decoding stages as a hierarchical symmetric U-shaped network. The real and imaginary domains of the audio are denoised separately, with the loss between $f_{\theta }\left(p_{1}\left(x\right)\right)$ and $p_{2}\left(x\right)$ being calculated to update network weights.

### 4.1. Voice Dataset Synthesis

To enhance adaptability to complex environments with noisy human voices, as shown in Figure 3b, two training datasets are generated via audio synthesis, covering *multiple scenarios* and *overlapping voices*:**Multi-scenario dataset**: To synthesize audio in a realistic voice dataset across various environments, we employ PyDub (https://pydub.com/, (accessed on 25 August 2025)) to overlap diverse UrbanSound8K (US8K) [39] noise onto clean audio signals from the Voice Bank dataset [40], which includes 28 speakers for training and 2 for testing. The complete noisy speech sample is generated by truncating or repeating the noise to cover the entire voice segment.**Multi-speaker dataset**: Using the same methodology as for generating the multi-scenario dataset, we combine the TIMIT dataset [41], which contains recordings of 630 speaker utterances, with the Voice Bank dataset [40] to generate an audio dataset of overlapping speaker voices. However, direct training on multi-speaker audio fails to achieve effective noise reduction performance. Hence, we employ Kullback–Leibler Divergence [42] to post-process the synthesized audio, thereby generating training data that can be effectively utilized in real-world HRI scenarios (more details in Appendix C).

### 4.2. Model Encoding and Decoding Stages

**Voice Encoder**. The encoding stage consists of four consecutive encoders for multi-scale voice spectral feature extraction. Each encoder’s downsampling block contains a complex convolution layer with a complex batch normalization layer (CBN) and a multilayer perceptron (MLP). It employs a complex filter W=(A+iB). For a complex vector h=c+id, it performs complex convolution via two independent real convolutions formulated as W∗h=(A∗c−B∗d)+i(B∗c+A∗d).

**Voice Decoder**. The decoder and encoder are symmetric architectures. Each decoder’s upsampling block contains a complex transposed convolution layer, followed by CBN and nonlinear MLP [3]. The decoding stage focuses on reconstructing the spectrogram locally and globally using voice features.

### 4.3. Self-Supervised Denoising Strategy

We illustrate the optimization function of the self-supervised strategy in Section 3.2, which incorporates the audio spectrum constraint. To prove the feasibility of the optimization problem, the *function rationalization* and *solution existence* of Equation (Equation 7) are analyzed.

**Non-negativity of the Objective Function.** For any θ, the squared Euclidean norm is non-negative; since expectation is the integral of the objective function, each term of the objective function is non-negative. We set(8)$$\left\{\begin{array}{c} L_{1}\left(\theta \right)=E{∥f_{\theta }\left(p_{1}\left(x\right)\right)-p_{2}\left(x\right)∥}_{2}^{2},\\ L_{2}\left(\theta \right)=E{∥f_{\theta }\left(p_{1}\left(x\right)\right)-p_{2}\left(x\right)-p_{1}\left(f_{\theta }\left(x\right)\right)+p_{2}\left(f_{\theta }\left(x\right)\right)∥}_{2}^{2},\\ L_{3}\left(\theta \right)=E{∥\log\left(S\left(f_{\theta }\left(p_{1}\left(x\right)\right)\right)\right)-\log\left(S\left(s_{2}\left(x\right)\right)\right)∥}_{2}^{2}. \end{array}\right.$$

Then, the function expectation is expressed as follows:(9)$$\left\{\begin{array}{c} \int_{x,y}{∥L_{1}\left(\theta \right)∥}_{2}^{2}f_{X,Y}\left(x,y\right)\;dxdy\geq 0,\\ \int_{x,y}{∥L_{2}\left(\theta \right)∥}_{2}^{2}f_{X,Y}\left(x,y\right)\;dxdy\geq 0,\\ \int_{x,y}{∥L_{3}\left(\theta \right)∥}_{2}^{2}f_{X,Y}\left(x,y\right)\;dxdy\geq 0, \end{array}\right.$$
where $f_{X,Y}\left(x,y\right)$ represents the joint probability density function of two-dimensional random variables *x* and *y*. The objective function $L\left(\theta \right)=L_{1}\left(\theta \right)+\mu L_{2}\left(\theta \right)+\delta L_{3}\left(\theta \right)\geq 0$, where δ≥0.

**Existence of Solutions within the Parameter Space**. The particular parameter space Ω is assumed to be a compact set (bounded and closed) in $R^{n}$. Since $f_{\theta }$ is continuous with respect to θ (*a differentiable function is continuous*) and $p_{1}$ and $p_{2}$ are measurable functions, by the property of the composition of continuous functions. There exist(10)$$\left\{\begin{array}{c} g_{1}\left(\theta ,x\right)={∥f_{\theta }\left(p_{1}\left(x\right)\right)-p_{2}\left(x\right)∥}_{2}^{2},\\ g_{2}\left(\theta ,x\right)={∥f_{\theta }\left(p_{1}\left(x\right)\right)-p_{2}\left(x\right)-p_{1}\left(f_{\theta }\left(x\right)\right)+p_{2}\left(f_{\theta }\left(x\right)\right)∥}_{2}^{2},\\ g_{3}\left(\theta ,x\right)={∥\log\left(S\left(f_{\theta }\left(p_{1}\left(x\right)\right)\right)\right)-\log\left(S\left(s_{2}\left(x\right)\right)\right)∥}_{2}^{2}. \end{array}\right.$$

Suppose that $f_{\theta }$ is bounded, i.e., there exist constants $C_{1}$ and $C_{2}$ such that $|f_{\theta }\left(p_{1}\left(x\right)\right)|\;\leq C_{1}$ and $|p_{1}\left(f_{\theta }\left(x\right)\right)|\;\leq C_{2}$ for all θ∈Ω.

It can be shown that there exist integrable functions $M_{1}\left(x\right)$ and $M_{2}\left(x\right)$ such that $|g_{1}\left(\theta ,x\right)|\leq M_{1}\left(x\right)$ and $|g_{2}\left(\theta ,x\right)|\leq M_{2}\left(x\right)$. Since logarithmic operations exhibit continuity within their respective domains, $g_{3}\left(\theta ,x\right)$ is continuous in θ, and there exists an integrable function $|g_{3}\left(\theta ,x\right)|\leq M_{3}\left(x\right)$ for all θ∈Ω.

Hence, the expectations $L_{1}\left(\theta \right)$, $L_{2}\left(\theta \right)$, and $L_{3}\left(\theta \right)$ are continuous in θ. Therefore, $L\left(\theta \right)=L_{1}\left(\theta \right)+L_{2}\left(\theta \right)+\delta L_{3}\left(\theta \right)$ is continuous in θ. According to the Lebesgue dominated convergence theorem [43], on the compact parameter space Ω, there always exists $\theta^{*}\in \Omega$ such that $L\left(\theta^{*}\right)=minL\left(\theta \right)$.

### 4.4. Denoising Network

Conventional 2D convolution kernels with fixed geometries usually struggle to capture detailed features in complex spectral data, particularly in sub-sampled voice signals. As formulated in Equations (1) and (2), directly separating the target voice from the noisy signal M(z) in the noisy environment is challenging.

To address this limitation, separating the real and imaginary components of the audio spectrum enables the denoising model to optimize magnitude and phase independently. Audio processing without component separation fails to establish a clear phase spectrogram structure [44], which is essential for perceptual quality in voice denoising. Hence, we aim to estimate the real and imaginary spectrograms of the target voice from the noisy signal via complex spectral mapping, simultaneously capturing *signal structure* and preserving *phase information*.

Therefore, our goal is to separately denoise the information in different dimensions (*real and imaginary*) to enable more effective extraction of audio features. The denoising network model π is composed of a convolutional multi-head attention architecture followed by a ResNet module (shown in Figure 5). Taking the real part $X_{r}$ and imaginary part $X_{i}$ of the voice spectral data as input, the model produces a complex-valued output *Y* defined as(11)$$Y_{r}=\pi \left(X_{r}\right)-\pi \left(X_{i}\right),$$(12)$$Y_{i}=\pi \left(X_{r}\right)+\pi \left(X_{i}\right),$$
where $\pi (X_{r})$ and $\pi (X_{i})$ represent the real and imaginary parts of the denoising model, and $Y=Y_{r}+jY_{i}$ denotes the model’s complex value output. This approach enables precise processing and reconstruction of spectrogram amplitude and phase data, while preserving voice contextual details. It improves the ability of robots to accurately understand key instructions during interactions (more details in Section 4.6).

### 4.5. Training Loss

Our primary objective is to satisfy the constraints of sub-sampled audio pairs by solving the optimization function in Equation (Equation 7) for voice denoising. Hence, the loss function consists of $L_{\mathrm{basic}}$ and $L_{\mathrm{reg}}$.

**Basic Loss**. We define the basic loss $L_{\mathrm{basic}}$ [45] as a combination of the waveform loss and an STFT loss to fit the target audio waveform:(13)$$\begin{array}{cc} L_{\mathrm{basic}} & =\frac{1}{TF}\sum_{t=0}^{T-1}\sum_{f=0}^{F-1}\left(|S_{r}\left(t,f\right)|\;+\;|S_{i}\left(t,f\right)|\right.\\ \; & \;\left.-|{\hat{S}}_{r}\left(t,f\right)|\;-\;|{\hat{S}}_{i}\left(t,f\right)|\right)\\ \; & \;+\frac{1}{N}\sum_{i=0}^{N-1}{\left(n_{i}-{\hat{n}}_{i}\right)}^{2}, \end{array}$$
where $n_{i}$ and ${\hat{n}}_{i}$ denote the *i*-th sub-sampled audio pair and its denoised version, respectively, with *N* being the total number of speech samples. Let *S* and $\hat{S}$ be the spectrograms of the sub-sampled and its denoised version, respectively, where *T* represents the number of frames and *F* represents the number of frequency bins.

**Regularization Loss**. We also need to consider the regularization loss discussed in Section 3.2. The regularization loss $L_{\mathrm{reg}}$, formulated as $L_{\mathrm{reg}}=L_{\mathrm{reg}1}+L_{\mathrm{reg}2}$, is defined as follows: (14)$$L_{\mathrm{reg}1}={∥f_{\theta }\left(p_{1}\left(x\right)\right)-p_{2}\left(x\right)-\left(p_{1}\left(f_{\theta }\left(x\right)\right)-p_{2}\left(f_{\theta }\left(x\right)\right)\right)∥}_{2}^{2},$$(15)$$L_{\mathrm{reg}2}={∥\log S\left(f_{\theta }\left(p_{1}\left(x\right)\right)\right)-\log S\left(s_{2}\left(x\right)\right)∥}_{2}^{2}.$$

The total loss *L* is the sum of the basic loss and the regularization loss, as $L=L_{\mathrm{basic}}+L_{\mathrm{reg}1}+\delta L_{\mathrm{reg}2}$. The performance of the model can be enhanced by adjusting the value of the parameters δ (for more details see Section 5.3).

### 4.6. Human–Robot Voice Interaction System

In this work, voice recognition is established as a precondition for robot task execution during HRI. As described in Section 4.4, the denoising model takes noisy speech *x* as input and outputs audio *y*, which contains the task instructions for guiding robot action. It enables the robot to more effectively identify key semantics in interaction by reducing background noise and overlapping voices. Based on the action information (i.e.,  *pick*, *grasp*, *pull*), the system executes various actions to interact with the environment. We develop an interaction system that integrates **voice recognition** and **action execution**. The VLA model is employed to generate robot action, with denoising voice serving as both the supporting condition for language prompts and the prerequisite for action execution.

An example is shown in Algorithm 1. When the interaction command is “Grasp the steel pipe and place it on the table behind”, the robot identifies the key semantics of the task instruction in the voice:**Grasp**: target task action;**Steel Pipe**: target object;**Table Behind**: placing position.

This approach provides sufficient task instructions and action information. Additionally, it offers an efficient approach to verify whether the audio has been effectively denoised. If the key semantic information within the denoised voice remains difficult to discern, the robot will be unable to execute the task action. Meanwhile, to enhance edge-case handling, we design action termination prompts (i.e.,  *stop*, *no*, *wait*). They can instantaneously terminate inference actions during HRI, prompting the robot to await subsequent task instructions (see Section 5.5 for a detailed discussion of our test).
**Algorithm 1** Human–robot voice interaction    **Input:** noisy voice
    **Output:** robot action
    RobotTask(noisy_voice)         interaction_audio ← **Denoise**(noisy_voice)         instruction ← **RecognizeVoice**(interaction_audio)         action ← **VLAModel**(instruction)         **return**
**RobotExecute**(action)


## 5. Experiment

In Section 5.2, we compare the denoising ability and generalization results of our method with previous self-supervised methods in simulation environments. The effectiveness of each component is validated in Section 5.3. In Section 5.4, we present application results of our method in real-world scenarios, including single-arm, dual-arm, and humanoid robot tasks. The effectiveness of termination prompts is tested in Section 5.5, demonstrating they can achieve closed-loop voice interaction control.

### 5.1. Experiment Setup

**Dataset**. Noisy signals from three distinct environments are used to evaluate algorithm robustness under varied conditions. In addition to the synthetic dataset introduced in Section 4.1, clean speech is augmented with white Gaussian noise to generate a synthetic white-noise training dataset. The training set includes 6000 white-noise voices as *Env1*, randomly selects 10,000 voices from the multi-scenario dataset as *Env2*, and takes 8000 voices from the multi-speaker dataset as *Env3*.

**Training Details**. In the denoising model, most convolutional layers adopt a 3 × 3 kernel and a 2 × 2 stride, while the middle two layers in the encoder and decoder utilize a 2 × 1 stride. We employ an Adam optimizer with a learning rate of 0.001, and the complex-valued voice spectrograms for model input are obtained via STFT with a 64 ms Hamming window and 16 ms hop size. All experiments were conducted on a workstation equipped with an Intel Core i9-14900K CPU and an NVIDIA RTX 4090 GPU for both data training and evaluation.

### 5.2. Evaluation in Simulation

**Metric**. To evaluate the denoising effect quality, we use five metrics in the simulation experiment [46,47,48]: signal-to-noise ratio (SNR), segmental signal-to-noise ratio (SSNR), wide-band perceptual evaluation of speech quality (PESQ-WB), narrow-band perceptual evaluation of speech quality (PESQ-NB), and short-term objective intelligibility (STOI).

**Baselines**. To comprehensively evaluate our model, comparisons are made with state-of-the-art baselines in voice denoising and self-supervised learning, including NCT [11], NNT [11], NerNT [10], and ONT [12]. These methods use noisy voice for model training and testing in the Env1-Env3 datasets. For more details on the differences in the training processes of the methods, see Appendix D.

**Result**. To ensure fair experimental evaluation, all baseline models are pre-trained on the same synthetic datasets. Table 1 shows that our model has achieved notable improvements in different environments and outperformed other baselines. Moreover, it has the ability to denoise in multi-speaker overlapping scenarios. This is because **(1)** the scaling dataset enhances self-supervised denoising performance across diverse scenarios and speaker backgrounds, and **(2)** the convolutional multi-head attention network improves feature extraction from discrete time–frequency representations (e.g., real and imaginary components).

We validate the model’s effectiveness by comparing its denoising performance with ONT [12] using a real-world utterance. Figure 6 presents the spectral changes before and after model processing. As shown in Figure 6a, the original interactive voice is heavily corrupted by noise, making it difficult to directly discern speech features (annotated in red). From Figure 6b,c, it is evident that the proposed method achieves significantly better performance in audio denoising. Figure 6c retains more command audio harmonics with less distortion than Figure 6b, demonstrating our method’s superior performance in a real-world scenario.

**Generalization Evaluation**. We evaluate the model’s generalization ability in unseen scenarios and real-world multi-speaker voices. Table 2 presents the denoising results of PESQ-NB and SNR under different scenario conditions, using training and testing datasets sourced from *Env2*. The model is trained using interactive voices under industrial noise (i.e., factory environment) and tested using interactive voices with street music (i.e., street environment). The results indicate that the model can successfully adapt to an unseen scenario.

Table 3 presents the the PESQ-NB evaluation results for multi-speaker voices from Env3, where *k* denotes the number of speakers. When k=0 (signifying no overlapping noise), we select three values (0, 1, 2) for the experiment. The presented method maintains effective noise reduction as *k* increases, achieving the highest voice quality compared to other methods.

### 5.3. Ablation Study

**Analysis of Regularization Loss**. Here we analyze the sensitivity of the regularization loss of the model optimization function in Equation (Equation 7). The SNR evaluation results are shown in Table 4, with five values (0, 1, 2, 4, 10) selected for experiments. The results demonstrate that model performance degrades when δ exceeds 2. This is because excessive constraints induce critical information loss, thereby leading to a poorer denoising effect. However, when δ is zero, the model fails to bridge the gap between the audio sub-sampler and ground truth, causing training instability.

**Analysis of Network Module**. Next, we analyze the sensitivity of different modules: (a) spectrogram training without separating real and imaginary components (denoted as Net-1), (b) feature extraction using convolutions without the attention module (denoted as Net-2), and (c) feature extraction using only the attention module (denoted as Net-3). Table 5 presents the denoising results, with the full model demonstrating the best performance. Two key reasons are identified: **(1)** Decomposing voice into real and imaginary parts can denoise time–frequency and phase simultaneously, enhancing information extraction from discontinuous audio. **(2)** The convolutional attention module can effectively suppress overlapping noise and extract critical voice features.

### 5.4. Evaluation in Real World

**Hardware Setup**. We perform real-world experiments using various robots: the Elite robot arm (7 DOFs), the dual-arm UR robot (12 DOFs), and the Realman humanoid robot (14 DOFs). All robotic arms are equipped with 1 DOF grippers, and external microphones are used for voice interaction, as shown in Figure 7.

**Tasks and Metric**. As shown in Figure 7 (right), we designed different tasks to evaluate model performance in real-world noisy scenarios, testing basic actions (e.g., *pull*, *grasp*, *wipe*) through voice interaction. The robots would recognize critical information in noisy interactive voice and execute instructed actions. The action success rate, calculated across multiple experiments, served as the primary metric.

**Implementation Details**. Figure 7 (left) shows the interactive environment. We designed two experimental conditions to evaluate real-world model performance: **(1)** computer-simulated urban music environments (https://pixabay.com/sound-effects/search/city/, (accessed on 25 August 2025)), and **(2)** overlapping speech with interfering speakers during voice interaction. The microphone collects interactive voice, and the noise meter measures environmental noise. We consider 50–70 dB as normal noise and above 70 dB as high-frequency noise (https://www.epa.gov/archive/epa/aboutepa/epa-identifies-noise-levels-affecting-health-and-welfare.html, (accessed on 25 August 2025)), and select RRnoise [49], DeepFilterNet [50], and ONT [12] as baselines with frozen weights to provide the denoised audio containing action information to the robot.

**Result**. Table 6 presents the instruction recognition success rates of these strategies, indicating that our model exhibits excellent generalization capabilities across various scenarios and overlapping interactive voices with shorter response times. It achieves notable improvements in noisy and overlapping environments, particularly under *70–80 dB mixed-audio conditions*, outperforming DeepFilterNet [50] and ONT [12] by 15% and 7.5% in success rate, respectively. Moreover, it can recognize action information in a discrete noisy voice, as shown in Figure 8. The enhanced performance is attributed to the use of training data augmented with overlapping voices, improving denoising capability in multi-speaker scenarios through accurate modeling of mixed-speech discontinuity.

The proposed model is also compared with state-of-the-art LLMs. The result indicates that Llama-Omni [15] handles ambient noise more effectively in a musical environment, demonstrating stronger robustness. However, its performance declines significantly in a multi-speaker scenario, whereas the proposed method achieves higher recognition success rates under the tested conditions. The performance degradation may occur because the multi-speaker conversational audio in its training data is out of distribution (OOD), making it difficult for the reasoning chain to infer target instructions. Notably, our model contains eight hundred thousand more parameters, which is relatively minor when compared to an 8B-parameter LLM backbone [5], while achieving shorter inference time for audio recognition.

### 5.5. Terminate Action Test

To verify the feasibility of interruptible interaction, we tested task-action changes through termination prompts during the interaction, as shown in Figure 9. The robot first executes the instruction “Pick the orange beside the apple”. At t=7 s, the “Wait” interrupt stops the action, and the robot returns to the initial pose. When t=10 s, a new instruction “Grasp the yellow cup handle and move to the white mat” initiates a new task. Termination words and instructions allow us to adjust action information, achieving a closed-loop HRI. Our mind is focused on improving the safety of the execution of actions through interrupt words, allowing the model to handle execution actions in *unseen* environments in a controlled manner.

## 6. Conclusions

In this paper, we introduce a self-supervised strategy for voice denoising, enabling effective HRI in unseen noisy real-world environments. Specifically, our method generates multi-scenario noisy voice datasets through data synthesis, constructs an effective complex-valued denoising network, and recognizes critical action information from discrete mixed audio. With the rapid advancement of HRI, this study focuses on enhancing recognition efficiency for action commands in noisy environments.

**Limitations**. Due to hardware limitations, real-word tasks in a laboratory cannot fully replicate outdoor environmental sounds, posing challenges for practical algorithm application. Moreover, convolutional attention networks cannot effectively separate and transfer speech attributes (e.g., content, pitch, timbre) [19], limiting style-preserving denoising. In future research, we will adapt flow matching for discrete speech modeling and deploy it on the kid-sized humanoid robot for outdoor validation.

## Figures and Tables

**Figure 1 biomimetics-10-00603-f001:**
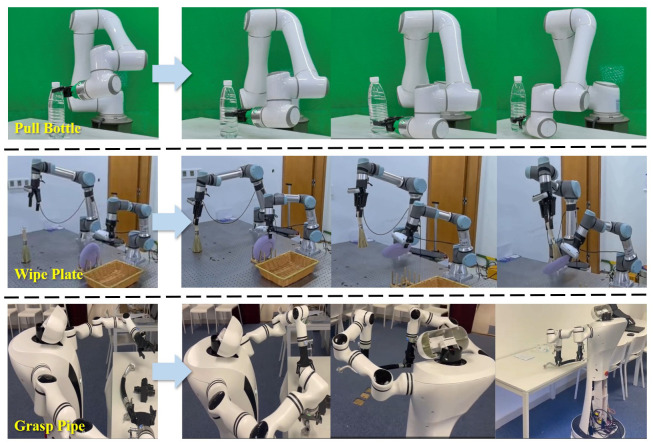
Three different tasks are executed by voice command. Their initial conditions are shown in the first column. The robot starts from an arbitrary position (second column), recognizing voice prompts in noisy environments and executing instruction actions (third and last column).

**Figure 2 biomimetics-10-00603-f002:**
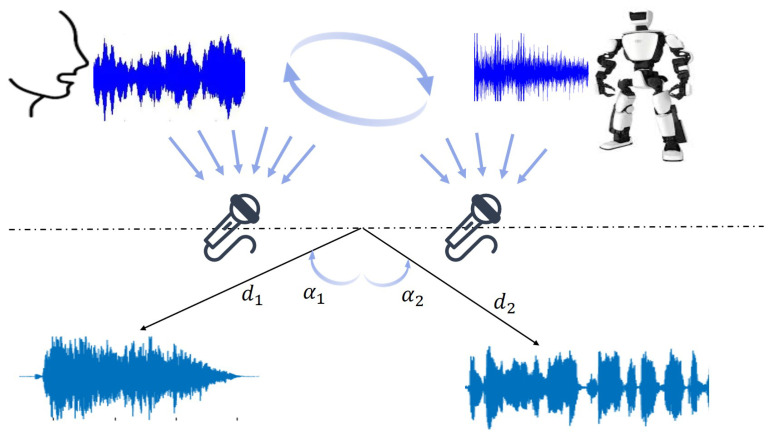
The process of blind source separation: contaminated speech picked up by multiple microphones first, and then the human speech and background noise are separated.

**Figure 3 biomimetics-10-00603-f003:**
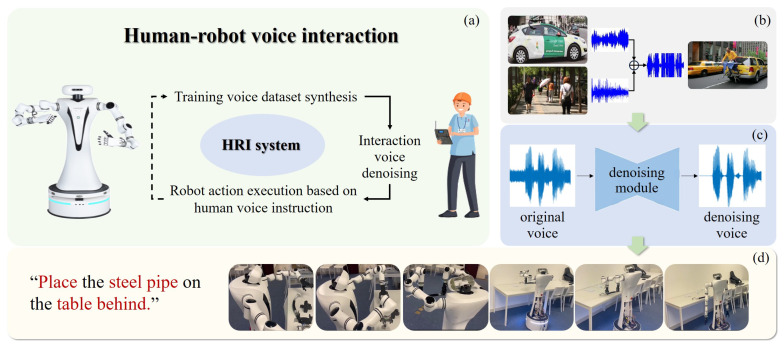
**Overview of HRI architecture.** (**a**) presents the system framework, which is componentized into three parts: (**b**) shows the data synthesis process, i.e., selecting two independent audio sources (car sounds and human voices) and fusing them to generate street audio. (**c**) indicates voice command denoising, and (**d**) demonstrates the robot executing actions based on interactive audio instructions.

**Figure 4 biomimetics-10-00603-f004:**
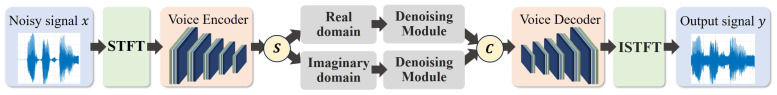
**Overall framework of the proposed self-supervised network.** During encoding, the network dynamically extracts multi-scale local and global features. The decoder then fuses these features to reconstruct voice spectral features. **S** separates signal frequency spectrum; **C** connects real and imaginary domains.

**Figure 5 biomimetics-10-00603-f005:**
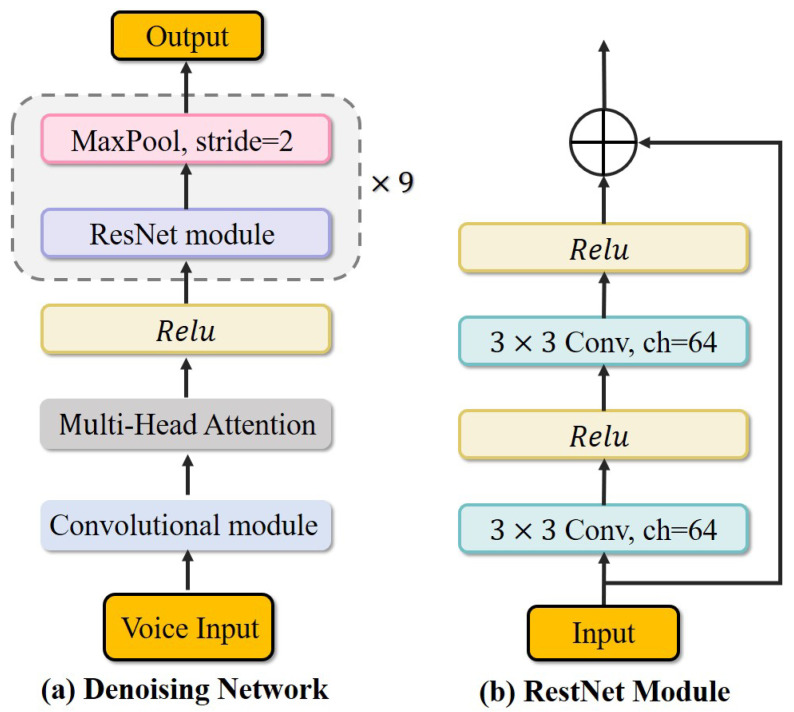
Detailed neural network architecture of the denoising module.

**Figure 6 biomimetics-10-00603-f006:**
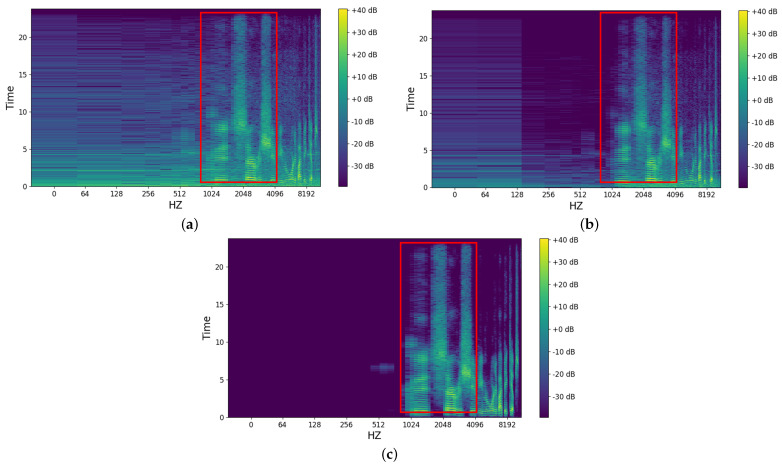
(**a**) displays the **original** noise signal spectrum. (**b**) illustrates the denoising output of the **ONT** method, and (**c**) depicts the output of **our** proposed method. Audio denoising should preserve the command audio signal (**annotated in red**) while eliminating background noise.

**Figure 7 biomimetics-10-00603-f007:**
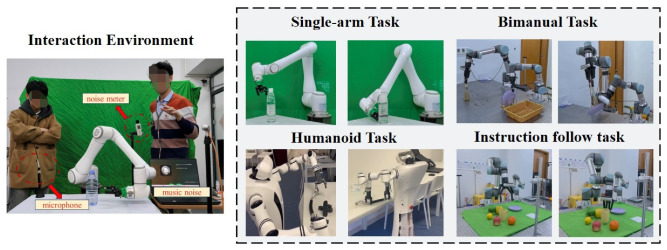
**Robot setup and examples for real-world manipulation tasks.** We validate the application of our algorithm on different robotic tasks.

**Figure 8 biomimetics-10-00603-f008:**
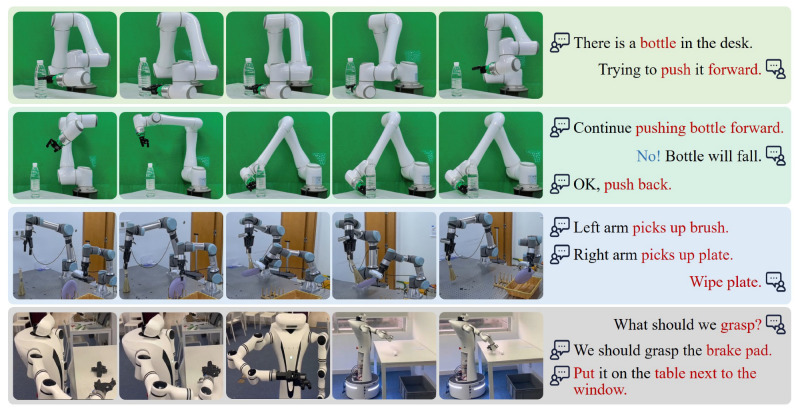
**The results of multi-turn voice interaction tasks.** Task instructions (red) and termination prompts (blue) are used to guide and adjust robot actions through discrete dialogues.

**Figure 9 biomimetics-10-00603-f009:**
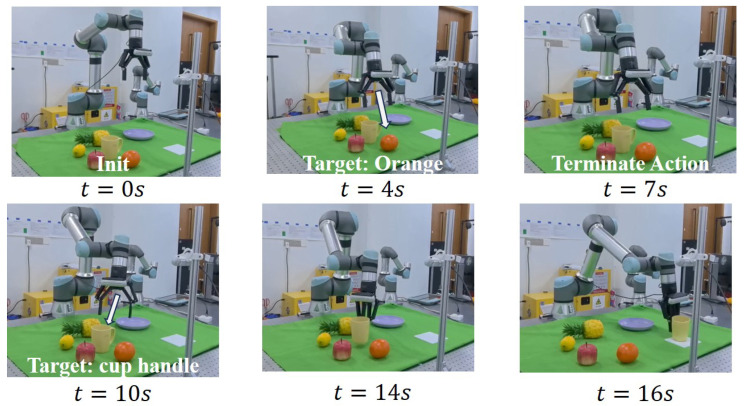
**Termination prompt application task.** During initiation, the audio command “Pick the orange beside the apple” is transmitted. Upon execution start, the current action is aborted via “wait” (t=7 s) and a new command is provided (t=10 s). The UR arm moves to the new target object, achieving closed-loop voice interaction control.

**Table 1 biomimetics-10-00603-t001:** Voice denoising evaluation with different methods.

Dataset	Network	SNR	SSNR	PESQ-NB	PESQ-WB	STOI
Env1	NCT	17.323±3.488	4.047±4.738	2.655±0.428	1.891±0.359	0.655±0.18
	NNT	16.937±3.973	3.752±4.918	2.597±0.462	1.943±0.375	0.645±0.018
	NerNT	−	−	−	−	−
	ONT	17.563±2.596	8.389±2.961	2.690±0.347	1.878±0.293	0.833±0.066
	ONT + rTSTM	18.137±2.122	9.077±2.437	2.643±0.317	2.003±0.282	0.839±0.067
	ONT + cTSTM	18.209±2.095	9.088±2.222	2.811±0.288	1.997±0.276	0.847±0.09
	Ours	18.671±3.093	9.628±3.002	2.961±0.508	2.166±0.326	0.864±0.073
Env2	NCT	3.44±3.457	−0.684±3.767	1.773±0.326	1.326±0.190	0.52±0.188
	NNT	3.99±5.451	−0.402±5.084	2.147±0.535	1.55±0.372	0.593±0.221
	NerNT	3.537±3.465	−1.336±3.105	1.787±0.26	1.249±0.126	0.569±0.177
	ONT	5.64±2.18	−0.28±0.69	2.15±0.189	1.76±0.39	0.54±0.56
	ONT + rTSTM	5.71±2.62	−0.23±1.36	2.23±0.27	1.83±0.35	0.68±0.27
	ONT + cTSTM	5.86±3.44	−0.15±3.6	2.41±0.304	1.97±0.21	0.72±0.019
	Ours	6.029±3.431	−0.074±0.993	2.818±0.556	2.123±0.26	0.87±0.23
Env3	NCT	3.75±3.899	−1.915±3.664	1.924±0.313	1.37±0.208	0.562±0.201
	NNT	4.428±3.166	−1.824±4.558	2.445±0.481	1.77±0.41	0.634±0.199
	NerNT	4.464±3.858	−1.837±3.714	2.321±0.351	1.484±0.256	0.651±0.198
	ONT	4.89±1.62	−1.52±2.45	2.58±0.317	1.58±0.273	0.662±0.16
	ONT + rTSTM	5.14±2.63	−1.45±2.36	2.61±0.17	1.69±0.31	0.674±0.18
	ONT + cTSTM	5.327±3.309	−1.391±4.22	2.65±0.49	1.84±0.324	0.69±0.21
	Ours	5.416±1.232	−1.223±2.458	2.727±0.437	1.991±0.189	0.712±0.16

**Table 2 biomimetics-10-00603-t002:** Generalization evaluation of unseen scenarios.

Training	Testing	Network	SNR	PESQ-NB
Factory	Street	NCT	3.645±3.676	1.8±0.46
		NNT	3.948±3.285	2.121±0.467
		NerNT	3.799±3.293	2.114±0.459
		ONT	4.38±2.31	2.33±0.39
		ONT + rTSTM	4.65±2.35	2.58±0.74
		ONT + cTSTM	4.91±2.46	2.73±0.4
		Ours	5.11±4.825	2.92±0.435

**Table 3 biomimetics-10-00603-t003:** Generalization evaluation of multi-speaker voices.

Network	*k* = 0	*k* = 1	*k* = 2
NCT	2.172±0.367	1.924±0.313	1.78±0.269
NNT	2.583±0.392	2.445±0.481	2.17±0.635
NerNT	2.56±0.26	2.321±0.351	2.13±0.29
ONT	2.63±0.16	2.58±0.317	2.24±0.439
ONT + rTSTM	2.68±0.21	2.61±0.17	2.29±0.524
ONT + cTSTM	2.71±0.23	2.65±0.49	2.38±0.15
Ours	2.802±0.376	2.727±0.437	2.41±0.29

**Table 4 biomimetics-10-00603-t004:** Ablation of regularization loss weights.

Noise	*δ* = 0	*δ* = 1	*δ* = 2	*δ* = 4	*δ* = 10
Env1	2.832	**2.961**	2.879	2.793	2.524
Env2	2.783	2.818	**2.837**	2.774	2.328
Env3	2.673	**2.727**	2.711	2.668	2.192

**Table 5 biomimetics-10-00603-t005:** Ablation of different network modules.

Dataset	Network	SNR	SSNR	PESQ-NB	PESQ-WB	STOI
Env1	Net-1	16.831	3.996	2.655	1.728	0.598
	Net-2	18.274	7.811	2.864	1.921	0.816
	Net-3	17.348	5.921	2.715	1.814	0.726
	Ours	**18.671**	**9.268**	**2.96**	**2.166**	**0.864**

**Table 6 biomimetics-10-00603-t006:** Voice denoising evaluation in real word.

Method	Urban Music (dB)		Multi-Speaker (dB)		Avg Time (s)
	50–70	70–80	50–70	70–80	
RRnoise	0.91	0.87	0.78	0.67	4.2
DeepFilterNet	0.94	0.91	0.82	0.72	1.9
ONT	0.92	0.90	0.84	0.79	3.1
Llama-Omni	0.98	0.97	0.86	0.82	3.7
Ours	0.96	0.94	**0.89**	**0.85**	**1.2**

## Data Availability

Data is contained within the article. Further inquiries can be directed to the corresponding author.

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
