# Peer review of "Self-Supervised Voice Denoising Network for Multi-Scenario Human–Robot Interaction"

_biomimetics, 2025, doi:10.3390/biomimetics10090603_

Round 1

Reviewer 1 Report

Comments and Suggestions for Authors
  1. The author should introduce the process of audio synthesis..
  2. Generate various datasets with minimal cost. How to evaluate the minimum cost?
  3. Check the correctness of Eq. (7).
  4. Check the correctness of Algorithm 1.
  5. Figure 6 needs more discussion and readability needs to be improved.

Reviewer 2 Report

Comments and Suggestions for Authors

Journal: Biomimetics (ISSN 2313-7673)

Manuscript ID: biomimetics-3730522

Type: Article

Title: Self-Supervised Voice Denoising Network for Multi-Scenario Human-Robot Interaction

Overall

The paper aims to address challenges in human-robot interaction (HRI) via voice commands, specifically focusing on two key limitations: the high cost of acquiring clean training datasets and the limited applicability of existing methods in complex real-world scenarios. The authors propose a method that leverages synthetic data and a self-supervised denoising network, with the goal of improving the real-world applicability of robotics.

Comments#

Comment 1 

(1) The statement in the abstract, “remains underexplored,” is misleading, given the considerable progress in voice-command HRI in recent literature (e.g., Kim, M.J., Pertsch, K., Karamcheti, S., et al. (2024). OpenVLA: An open-source vision-language-action model. arXiv:2406.09246, Brohan, A., Brown, N., Carbajal, J., et al. (2023). RT-2: Vision-language-action models transfer web knowledge to robotic control. arXiv:2307.15818). Please revise to acknowledge prior work and specify what unique aspects (e.g., robustness to overlapping noise, deployment with synthetic data) are still lacking. (2) Additionally, “complex real-world scenarios” is too vague. Please define what constitutes “complexity” in your context (e.g., environmental noise levels, speaker overlap, multi-turn dialogues), provide concrete benchmarks, and compare with state-of-the-art methods. (3) The claim that your method “outperforms other training strategies in simulation benchmarks” should also clarify whether the main contribution is in synthetic dataset generation or the self-supervised denoising network architecture.  I recommend revising the abstract to be more concise, with a clear and logical flow, specifying both the technical novelty and the practical impact, and grounding claims in comparison to recent literature.

Comment 2

Figures 1 and 2 currently appear in the introduction, but Figure 2 especially is discussed in Section 4 (Methodology). For clarity and better narrative flow, please consider relocating these figures to the Methodology section where they are first discussed in detail.

Comment 3

The contribution section can be condensed for clarity. The first two points— (1) training denoising models with synthetic data and (2) a novel self-supervised learning strategy—constitute the core technical contributions. The third and fourth points are outcomes or applications rather than standalone contributions and could be integrated into the Results/Discussion. Please revise the contribution statement accordingly for conciseness and impact. Please avoid the word of “Novel”

Comment 4

Your introduction discusses Vision-Language-Action (VLA) models, but recent progress in large language models (LLMs) and multimodal transformers for HRI (Y. Lai et al., "Natural Multimodal Fusion-Based Human–Robot Interaction: Application With Voice and Deictic Posture via Large Language Model," in IEEE Robotics & Automation Magazine, doi: 10.1109/MRA.2025.3543957, Kim, M.J., Pertsch, K., Karamcheti, S., et al. (2024). OpenVLA: An open-source vision-language-action model. arXiv:2406.09246) raises the question of whether LLM-based systems could bypass the need for synthetic audio and self-supervised denoising.

Please provide a discussion or experimental evidence comparing your pipeline to emerging LLM-driven approaches (e.g., voice-to-text-to-action) and clarify the added value of your contribution in this rapidly evolving context.

Comment 5

While the introduction provides some context for the topic, it could be improved in terms of structure and depth. A well-structured introduction should:

  • Clearly establish the background and importance of human-robot interaction via voice commands.
  • Summarize key recent advances in both supervised and self-supervised speech denoising, especially in HRI settings.
  • Highlight the specific limitations or open problems that remain unaddressed in the current literature, with appropriate citations to the most relevant and recent works.
  • Clearly articulate the novel contributions and significance of the proposed approach.

Currently, the introduction does not sufficiently review or synthesize the state of the art, and it does not clearly outline how this work builds upon or differentiates from existing methods. I recommend restructuring the introduction to include a more comprehensive literature review, clearer problem definition, and an explicit statement of your contributions in relation to previous work. This will help readers better appreciate both the context and the value of your proposed method.

Comment 6

The field of human-robot interaction using voice commands is rapidly evolving, especially with the integration of large multimodal and language models (OpenVLA,2024). Please expand the related work and discussion sections to compare your method more directly with these recent advances. Explicitly discuss the circumstances under which your self-supervised denoising method offers unique advantages, or where it could be integrated with LLM-based pipelines.

Comment 7

While your manuscript compares the proposed self-supervised denoising network with several baselines, I notice that the chosen baselines (e.g., NCT, NNT, NerNT, ONT) are either traditional or partially related to speech denoising and do not fully represent the current state of the art in deep learning-based speech enhancement. For instance, you included “RRnoise” (Valin, 2018) and DeepFilterNet (Schroter et al., 2022), which focus on real-time, low-complexity full-band speech enhancement using hybrid DSP/deep learning and deep complex filtering, respectively. However, these methods—while strong in real-time and low-resource contexts—do not specifically target the HRI scenario or leverage self-supervised or synthetic data as in your work.

More critically, your manuscript does not compare against several recent methods that are directly relevant to your target problem of self-supervised speech denoising for human-robot interaction, such as:

  • Wu et al. (2021): “Self-Supervised Speech Denoising Using Only Noisy Audio Signals”
  • Xu et al. (2020): “Listening to Sounds of Silence for Speech Denoising”
  • Li et al. (2024): “Single-Channel Robot Ego-Speech Filtering during Human-Robot Interaction”

These works focus specifically on self-supervised learning for speech denoising in robotic or interactive contexts, closely aligning with your manuscript's aims. Some of these models also address the challenges of training without clean datasets, like your approach, and are thus highly relevant as direct baselines.

Can you clarify why you have not provided comparisons with these more recent, highly relevant methods? Including such comparisons, or at minimum a detailed discussion contrasting your approach with these methods, would greatly strengthen the scientific rigor and contextual relevance of your work. It would also allow readers to better appreciate the unique advantages and contributions of your proposed model within the rapidly evolving field of speech enhancement for HRI.

Comment 8

For real-world adoption, practitioners need to understand the computational requirements and practical integration steps. Please provide more details on the computational costs (training and inference time, hardware requirements) and discuss any challenges in deploying your approach on different robotic platforms or with off-the-shelf hardware.

Comment 9

While the manuscript briefly discusses limitations in Section 6, please expand this section to include a more thorough discussion of potential failure cases, boundary conditions (e.g., types of noise not handled), and situations where your approach may underperform compared to alternatives. Such transparency aligns with best practices in AI research (see NeurIPS Ethics Guidelines).

Comment 10

The manuscript would benefit from careful language editing to improve clarity, grammar, and readability. There are several instances of awkward phrasing and grammatical mistakes throughout the text that reduce the clarity of the scientific content and may hinder understanding. For example, in the abstract: “Two limitations remain as resolved such as the high cost of acquiring clean training datasets and the limited applicability in complex real-world scenarios.” This sentence is confusing and should be rephrased for clarity. I recommend a thorough review of the manuscript for grammatical errors, sentence structure, and academic style, potentially with the assistance of a professional English editing service. Improving the language quality will help ensure that the technical contributions of your work are communicated more effectively to a broad audience.              

Reviewer 3 Report

Comments and Suggestions for Authors

This paper has some merit; however, I find that the paper has two major flaws.
First, the ultimate experiment is to have a robot outdoors (with significant background from vehicles and pedestrians) or at a function (where there are many conversations simultaneously going on). Such real-world environments are typically tested in the humanoid league of RoboCup, and they provide the environmental conditions that this research aims to address.  Experiments under the setting of Figure 7 seem particularly limited, as there are very few sources of external noise, and the setting corresponds to laboratory conditions and not real-world situations, even for the mobile humanoid robot.   Second, the paper is particularly poorly written. The goal of publications is to communicate the entire process as clearly and transparently as possible to let the community reproduce the results faithfully. I am unable to reproduce the results here, simply because the processes and algorithms, as well as the experiments, are described with a lack of detail and with poor organisation. I am unable to grasp how much noise is added when it is added. Are the sounds played back on speakers and captured jointly in a new microphone or is the merging performed by software tools? What parameters are involved, and what are their ranges? Why are the authors not offering to release the datasets before and after the introduction of noise? Can the methods applied be described by presenting a data-flow diagram and numbering each step? Can the mathematical justification of the methods be derived more clearly?    

Comments on the Quality of English Language

The English description needs improvement. While plurals such as “works,” “noises,” and “expenses” are valid words in English, the usage here is technically incorrect; the authors mean “different types of noise,” not just “noises,” or “related work,” not “related works.” Why "environmental interferences" and not "environmental interference"? Why the plural?

There are too many confusing semantic issues caused by the imprecise use of English, making the paper almost unreadable. For example, “supervised training networks” likely meant “training networks under supervised learning,” and “How can we scale training data to simulate audio information in real-world multiple scenarios?” should be “How can we scale training data to simulate audio information in real-world multiple-SPEAKER scenarios?”

Mathematics is also English prose. Use punctuation. For example, Equation (2) and Equation (3) are missing a full stop.

Round 2

Reviewer 2 Report

Comments and Suggestions for Authors
  1. Abstract: Please report at least one numerical result in the abstract to substantiate the contribution. This will help readers quickly grasp the effectiveness of your approach.
  2. Introduction: The third listed “contribution” is not a contribution per se; it is a performance assessment. Either remove it from the contribution list or reframe it as: “We demonstrate proof-of-concept of the proposed approach in both simulation and hardware.”
  3. Move Figure 2 to the Methodology It appears to depict the system/framework and is more appropriate there than in the Introduction. Keeping Figure 1 in the Introduction is sufficient to orient the reader. Also, Figure 2 is not discussed in the Introduction.
  4. Remove references such as “as shown in Fig. 2” from the Related Work section. Readers do not need your framework figure to understand prior art. Suggested rewrite: “In our work, we improve the model’s denoising capability for unseen environments by synthesizing audio from diverse scenarios and expanding the training dataset.” (No cross-reference to Fig. 2 is needed here.)
  5. Please add a subsection at the start of Methodology titled “Project Overview.” In this subsection: (1) Briefly describe the system workflow at a high leve, (2) Place Figure 2 here to illustrate the pipeline, and, and (3) Then proceed to the remaining subsections
